

# Estimation of hazard assessment by FINSIM for west coast and son narmada faults

Shivamanth Angadi[1], Mayank Desai[2]

[1]Research Scholar, Dept. of Applied Mechanics, SVNIT, SURAT-395007, India
[2]Assistant Professor, Dept. of Applied Mechanics, SVNIT, SURAT-395007, India

*Correspondence to*: Shivamanth Angadi (shiva05cv@gmail.com)

**Abstract.**The Seismic hazard study was carried out for Maharashtra state, Bombay (Latitude 18.940 N, Longitude 72.840E). In the present study the geological fault is known as West coast fault and Son Narmada Faults were studied and used for the earthquake simulation, extended finite fault method originally FINSIM given by M. Atkinson (1998), was used to simulate an earthquake of 6.5 Mw. The soil classification was carried out by the Shear wave velocity and the relation between Shear wave velocity and SPT valves was also recommended by Sumedh Y. Mhaske (2011), since the Mumbai has been formed by the conglomeration of various islands which has come together to form a single landmass. The soil investigation suggested that Most of the region comes under the Class D and C for the worst case simulation we have used the site class D. The peak ground accelerations (PGA) vary from 0.03g to 0.133g. While coming to zonal area IS1893:2002 still consider the Mumbai city under zone III with Z value of 0.16 and the result have been compared with the analysis done by many research in the same area.

## 1 Introduction

Now-a-days millions of people in the different part the world live with a significant risk to their lives and property from earthquakes. Billions of dollars of public infrastructure are continuously at risk of earthquake damage. These risks are not unique to the United States, Japan, or any other country. Earthquakes are global phenomenon and a global engineering problem. Some specific regions have been witnessing earthquakes repeatedly but it doesn't mean that some regions are earthquake free even the peninsular India so called stable region has witnessed the earthquakes of large magnitude. The following past earthquakes have made the government and the research institutes to work towards the disaster mitigation programme  or the disaster management program, September 30th 1993 khillari , May 22nd Jabalpur, March 29th 1999 and the famous and most disastrous Bhuj earthquake on 26th January 2001.

To understand the evolution of seismic hazard estimation or the peak ground acceleration estimation, mathematical theories were used in order to simulate or represent the seismic wave propagationby the past researchers. As per the present region of study is considered lot of work has been done by many researchers using deterministic and probabilistic seismic hazard





estimation, an attempt has been made in the present study to simulate earthquake in the West Coast and son narmada faults using Extended finite fault modeling or FINSIM program.

## 2 The Study Area and Geographical Details

The present research work the seismic sources within the radius of about 300km are considered with Mumbai city (latitude
18.940N, longitude 72.840E).  The control region covers latitudes from 16.00 to 22.000 N and longitudes from 69.90 to 75.900 E and includes major part of western Maharashtra, southern Gujarat and Union Territory of Daman. The considered area falls under zone II-IV. Mumbai and Daman (Union Territory also the southern part of Gujarat) lie in zone III, Warna and Koyna region on western side of Maharashtra lie in zone IV, rest of  rest of western side of Maharashtra within the control region falls in zone II and III.

## 3 Methodology

Finite source modelling is been an important stage of ground motion prediction process near the epicenter of the large earthquake. In most of the studies the entire fault is discretized into small elements, and each small element is treated as small source, and the radiation or the energy given or dissipated by all such sources is summed with the proper delay in time. The rupture begins at the given point or at hypocenter and propagates with constant velocity and it will trigger sub-sources as
soon as it reaches them. This simple kinematic phenomenon applied on all sub faults differ chiefly on how the path effects and sub-source spectra are defined. The sub-events which are modeled out of the faults are characterize as "stochastic" $\omega^2$ sources and the empirical distance dependent duration, attenuation models and geometric spreading are used to define and describe the path effects. This stochastic method has been successfully used to predict earthquakes that could be treated as point sources, the aim of the FINSIM or the EXSIM is to extend this technique to large faults and The amplifications can be
used directly in the stochastic model for simulating earthquake ground.

## 4 Fault Details and Earthquake Model Parameters

West coast fault have witnessed an earthquake of magnitude Mw 5.2 in the year 1993 also according to Raghukanth and Iyengar (2006), Seismic hazard of Mumbai region is controlled by west coast fault. As mentioned earlier the geometrical and geological details of both the faults are taken from seismotectonic map of India published by GIS. The geometrical
parameters are given in Table no 1, 2and 3.



## 5 Results and Discussion

Entire Mumbai was divided into the grids of size 002`0`` X 002`0`` Show in Fig 1 which divides the area into 5km by 5km square grid the Spectral accelerations and acceleration time history are evaluated at the intersection of the grid point there are about 69 points. Other than these 69 grid points the results are also calculate for the important site and areas in the Mumbai city. The coordinates of the important places and cities are given in Table 4 and 5 and the results of the peak ground acceleration and spectral acceleration at the important sites and places are tabulated in Table 6.

In Fig 2 (a) and (b) are the contour maps of PGA values due to West Coast fault at important sites and places which are given in Table 4 and 5. The central Mumbai is highly vulnerable, mainly the region from Chatrapathi Shivaji International Airport to Baba Atomic Research Centre (BARC). The PGA values of West coast fault with 0.035g to 0.059g, Sarika D and Chaoudary D, (2014), have carried out the seismic hazard analysis for the Mumbai city using deterministic and probabilistic methods, Fig 2 (c) and (d) shows the comparison of the contour maps of PGA values since the author have calculated the PGA values at bed rock level, by DSHA method PGA values vary from 0.17 to 0.626g and by PSHA method 0.17 to 0.37g since the site amplification factor are also considered in present work the PGA values vary between 0.36 to 0.565g. Fig 3 shows the graphs of Spectral acceleration Vs time at important site and place.

## 6 Conclusion

The present study is an attempt to simulate the earthquake at the geological faults near the study area using Extended Finite fault Method. The seismic hazard was first estimated at bed rock level and then with the incorporation of soil amplification the hazard was estimated at the surface. The peak ground accelerations (PGA) vary from 0.03g to 0.133g. (g) values for the entire grid points. Raghukanth and Iyengar also calculated the PGA (g) values for Mumbai region by PSHA method, PGA (g) value was found out to be 0.28 few sites with shear velocity between 760-1500 m/s, as we have considered the soft soil condition to have worst case scenario we have got higher value. Similarly spectral acceleration values vary from 0.1g to 0.44g for Class D site in our case the Sa/g values vary from 0.21g to 0.7g for the most of the class.

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

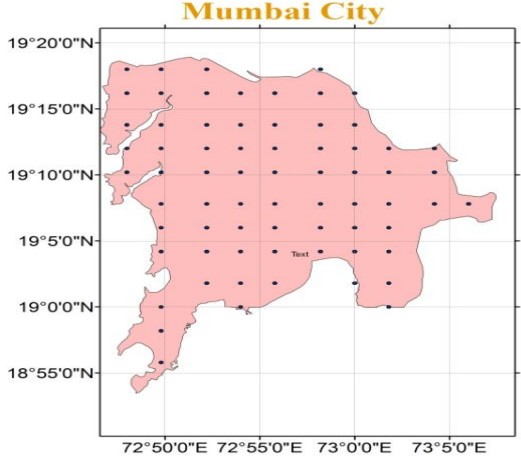

**Fig. 1: Mumbai city with grid points at 5kmX5km.**





**Fig. 2:  Contour maps of PGA values at important sites and places of West Coast Fault and Comparison.**





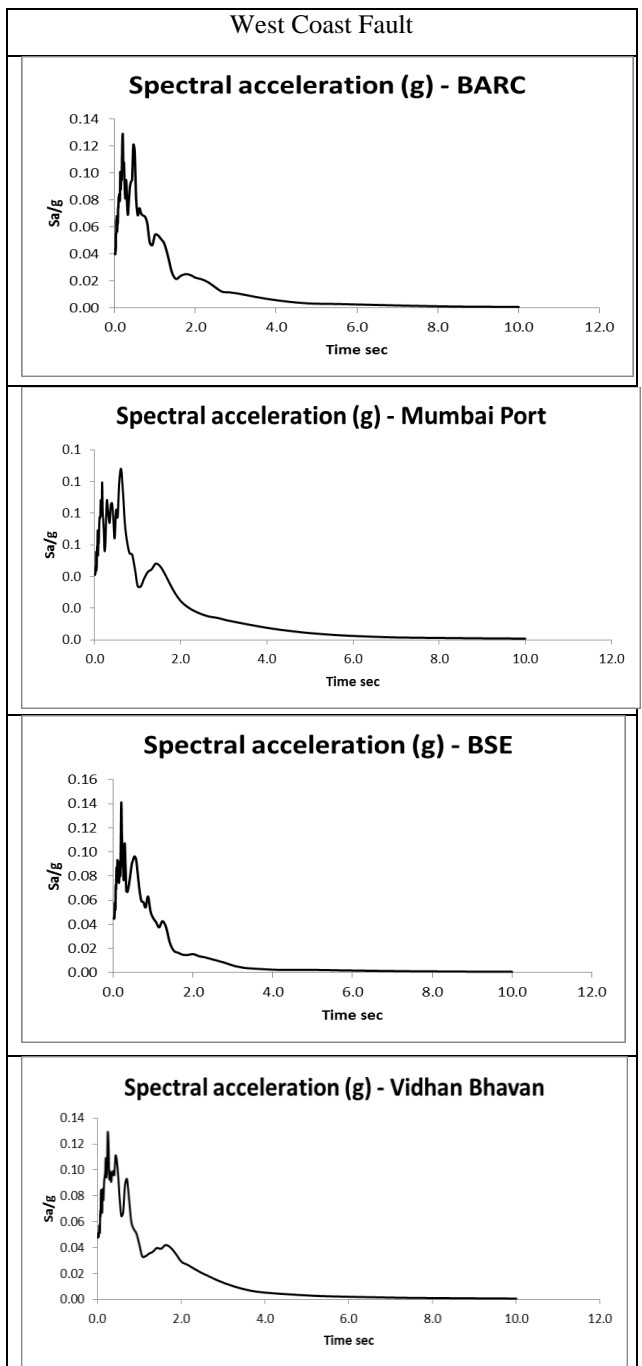



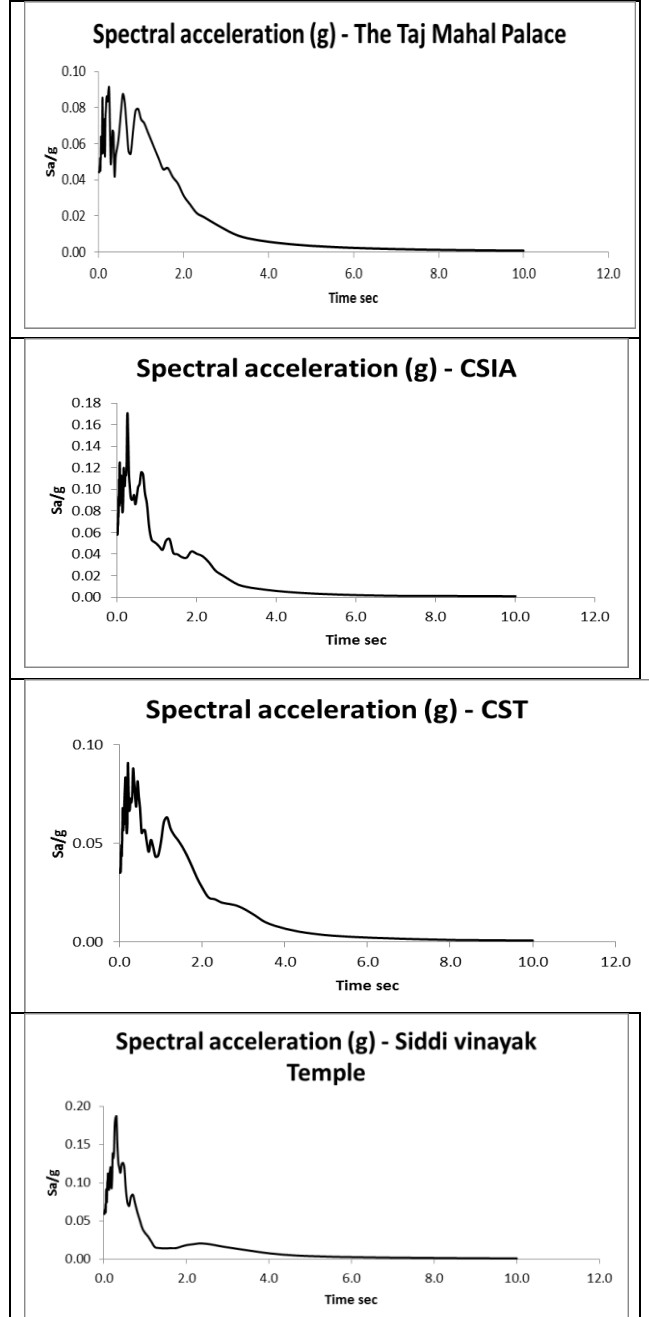





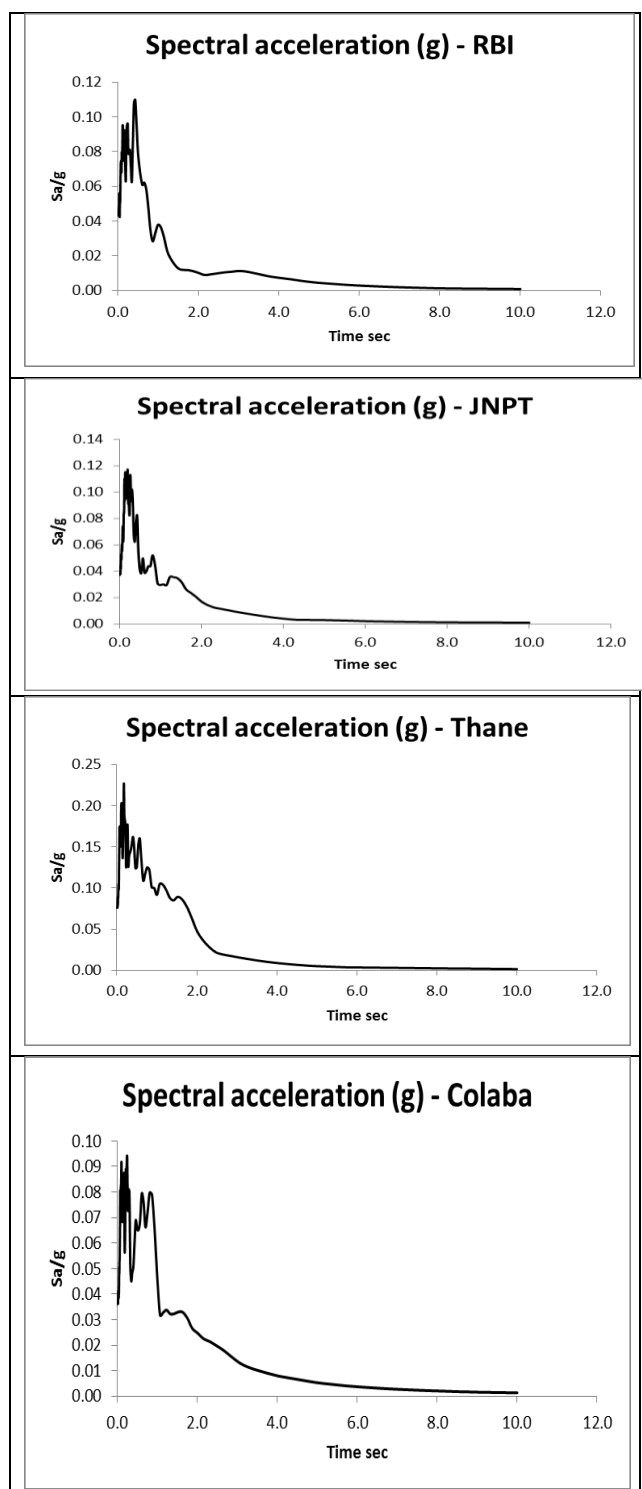





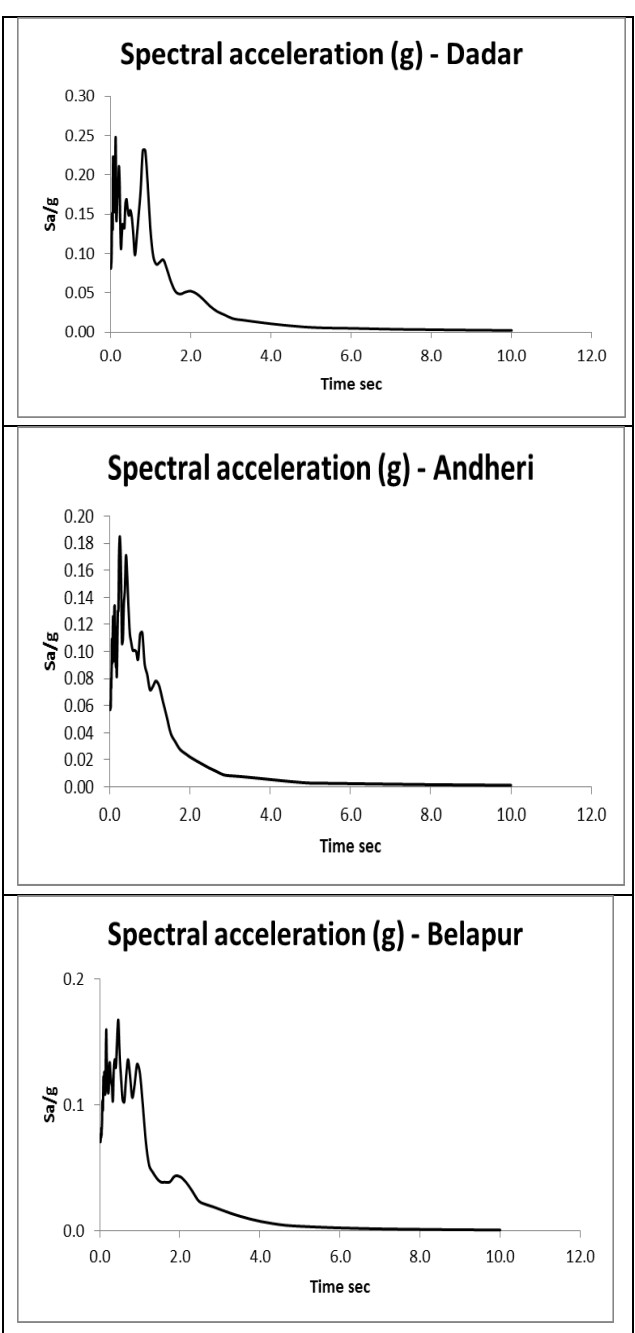

Fig 3. Spectral acceleration graphs



**Table : 1 Geometrical Fault details**

| Name of the fault | Start co-ordinate | | End co-ordinate | | Distance Km | Earthquake Mw |
|---|---|---|---|---|---|---|
| | Latitude | Longitude | Latitude | Longitude | | |
| West Coast | 19.99 | 72.85 | 18.008 | 73.20 | 370.5 | 5.2 |

**Table 2 :  Geological Fault details**

| Weast Coast Fault | |
|---|---|
| Earthquake date | 12-08-1993 |
| Location | Lat 17.03 Long 73.65 |
| Magnitude and depth | Mb-4.9, Depth-32 Km |
| Type | Np2 |
| Strike | $173^0$ |
| Dip | $58^0$ |

5    **Table 3 :  Model Parameters used for the simulation of ground motion in the present study for Weast coast fault**

| Magnitude | 6.5 | Present work |
|---|---|---|
| Latitude , Longitude | $19.99^0$ N, $72.85^0$ E | Present work |
| Hypocenter depth | 15 km | Present work |
| Fault strike Dip | $173^0$,$58^0$ | Present work |
| Fault length width | 25km,12km | Present work |
| Slip Distribution | Random | Present work |
| No faults along strike, dip | 25,12 | Present work |
| Location of the hypocenter on fault plane | 1,1 | Present work |
| Shear wave velocity | 3.6 km/s | Hard rock |
| Density | 2.8g/cm$^3$ | Hard rock |
| Stress drop | 140 | Present work |
| Kappa | 0.016 | Present work |
| Geometric Spreding | $1/R - R \leq 40 km$ | Bodin et al 2004 modified by Sumeer |





| | | chopra 2012 |
|---|---|---|
| | $1/R^{0.5} - (40 \leq R \leq 80km)$ | |
| | $1/R^{0.55} - (R \geq 80km)$ | |
| Duration properties | $f_c^{-1} - (R < 10km)$ | Atkinson and Boore (1995) |
| | $f_c^{-1} + 0.16R - (10 \leq R \leq 70km)$ | |
| | $f_c^{-1} + 0.04R - (130 \leq R \leq 1000km)$ | |
| Attenuation relation | $84f^{0.65}$ | B K Rastogi and A G Chhatre (2014)[4,9], |
| Pulsing Percent | 50 | |

**Table 4 : Important sites**

| Latitude | Longitude | site |
|---|---|---|
| 19.02 | 72.93 | BARC |
| 18.94 | 72.77 | Mumbai port |
| 18.93 | 72.83 | BSE |
| 18.93 | 72.82 | M vidhan bhavan |
| 18.92 | 72.83 | Taj mahal |
| 19.10 | 72.87 | CS intr airport |
| 18.94 | 72.84 | CST rail |
| 19.02 | 72.83 | Siddivinaayak Temple |
| 18.93 | 72.84 | RBI |
| 18.95 | 72.95 | JNPT |

**Table5 : Important places**

| Latitude | Longitude | Place |
|---|---|---|
| 19.21 | 72.97 | Thane |
| 18.91 | 72.81 | Colaba |
| 19.21 | 72.84 | Dadar |
| 19.11 | 72.87 | Andheri |





| 19.03 | 73.04 | Belapur |
|---|---|---|

**Table : 6 Peak ground acceleration and spectral acceleration values at important sites and places**

| Site | Latitude | Longitude | West coast Fault PGA(g) | West coast Fault Sa/g 0.2sec | West coast Fault Sa/g 1.0sec |
|---|---|---|---|---|---|
| BARC | 19.02 | 72.93 | 0.040 | 0.104 | 0.054 |
| Mumbai Port | 18.94 | 72.77 | 0.041 | 0.074 | 0.034 |
| BSE | 18.93 | 72.83 | 0.045 | 0.100 | 0.045 |
| M vidhan bhavan | 18.93 | 72.82 | 0.048 | 0.101 | 0.043 |
| Taj mahal | 18.92 | 72.83 | 0.044 | 0.080 | 0.074 |
| CS intr airport | 19.10 | 72.87 | 0.058 | 0.107 | 0.050 |
| CST rail | 18.94 | 72.84 | 0.035 | 0.069 | 0.051 |
| Sdiddivinaayak Temple | 19.02 | 72.83 | 0.059 | 0.118 | 0.034 |
| RBI | 18.93 | 72.84 | 0.043 | 0.084 | 0.038 |
| JNPT | 18.95 | 72.95 | 0.037 | 0.098 | 0.030 |
| Thane | 19.21 | 72.97 | 0.076 | 0.171 | 0.092 |
| Colaba | 18.91 | 72.81 | 0.036 | 0.081 | 0.047 |
| Dadar | 19.21 | 72.84 | 0.081 | 0.171 | 0.130 |
| Andheri | 19.11 | 72.87 | 0.057 | 0.123 | 0.072 |
| Belapur | 19.03 | 73.04 | 0.057 | 0.104 | 0.101 |