# Peer review of "Estimation of hazard assessment by FINSIM for west coast and son narmada faults"

_Natural Hazards and Earth System Sciences, 2018_

## Referee Comment (RC1) · Anonymous Referee #1 · 2 Mar 2018

The manuscript by Angadi and Desai (2018) deals with ground motion simulations in West Coast, India. I regret to state that the manuscript is not prepared with care as an academic paper and it does not add to existing literature. The amount of revisions required exceed reasonable amount of major revisions and thus it is rejected herein. Following are my specific comments:

1. The manuscript is full of grammar and language mistakes. There are so many mistakes that the manuscript is not comprehensible at several locations.

2. The language is not technical and the text does not stand as an academic paper. Many repetitions are there in the text. In addition to the language issues, regular rules

of academic papers are not followed. For instance: There are no citations in the text to the references, we only see them in the reference list.

3. The Introduction section does not cite the relevant past research on the topic, even though the strong motion simulations are used and methods are validated worldwide. All these studies are ignored in this manuscript.

4. There is no detailed information on the geology of the study area. Also, no information on the local conditions is provided.

5. The application details are not clear. Which event is modeled in the manuscript? Is it the 1992 Mw=5.2 event? The Mw value of the simulated event is not clear, we see Mw=5.2 in the text but see different values in the Tables 1-3 (Mw=6.6 in Table 3).

6. The simulations are not validated. Is there any dataset recorded from the mentioned earthquake? Or are the authors performing blind simulations in the study area? If they are simulating a real event, a comparison of simulated motions against data is necessary. In addition, comparisons against validated GMPEs is necessary. Otherwise, the entire simulation exercise is not verified.

7. The comparisons provided in Figure 2 are not comprehensible. The PGA levels in the top panel is very low while the lower panel has higher PGA values. We do not understand which panel represents what study? Why are the values so different? In summary, the simulations are not verified.

8. Authors compare PSHA and DSHA results at some point in the text. These two type of methods are not comparable at all.

9. The Conclusions also show that the study is one simulation exercise whose details are not presented and whose validation is not performed.

---

## Referee Comment (RC2) · Anonymous Referee #2 · 10 Mar 2018

The manuscript "Estimation of hazard assessment by FINSIM for the west coast and son narmada faults" by Shivamanth Angadi and Mayank I feel that the manuscript in present form can not be published in Nat. Hazards Earth Syst. Sci .. I suggest to reject it, considering that a great amount of work is actually necessary to improve the manuscript. Some general suggestions:

**Abstract**
- The abstract contains information such as the name of "zonal area" that international reader doesn't know. Thes information should specified in other parts of the manuscript.
- Usually, references are not quoted in the abstract.

**The Study Area and Geographical Details**
- This section needs a picture showing the quoted locations. Moreover, other information on the general seismic hazard (e.g. active faults, past earthquakes, density of inhabitants, … )

**Methodology**
- No information are given on source parameters, attenuation and site effects used. Although, in the abstract authors write about site effects estimate made using shear wave velocity measurements and SPT. Some data are in the table, but are quoted in the text in other sections.

**General comments**
- Several sections could be merged.
- English language need revisions.
- The manuscript seem to be an abstract. Some sections should be improved, adding more details.
- In the conclusions the authors should describe the novelty of their approach in seismic hazard computation.
- Figure captions should be improved adding details.
- Pictures need revisions, especially as concern their layout